# Vascular Lipidomic Profiling of Potential Endogenous Fatty Acid PPAR Ligands Reveals the Coronary Artery as Major Producer of CYP450-Derived Epoxy Fatty Acids

**DOI:** 10.3390/cells9051096

**Published:** 2020-04-29

**Authors:** Matthew L. Edin, Fred B. Lih, Bruce D. Hammock, Scott Thomson, Darryl C. Zeldin, David Bishop-Bailey

**Affiliations:** 1Division of Intramural Research, NIEHS/NIH, Research Triangle Park, NC 27709, USA; 2Department of Entomology and Comprehensive Cancer Center, University of California, Davies, CA 95616-8584, USA; 3Royal Veterinary College, University of London, London N1 0TU, UK; 4North Cornwall Research Institute, Bude, Cornwall EX23 9EE, UK

**Keywords:** PPARs, vascular, coronary artery, lipidomics, eicosanoids, inflammation, CYP450

## Abstract

A number of oxylipins have been described as endogenous PPAR ligands. The very short biological half-lives of oxylipins suggest roles as autocrine or paracrine signaling molecules. While coronary arterial atherosclerosis is the root of myocardial infarction, aortic atherosclerotic plaque formation is a common readout of in vivo atherosclerosis studies in mice. Improved understanding of the compartmentalized sources of oxylipin PPAR ligands will increase our knowledge of the roles of PPAR signaling in diverse vascular tissues. Here, we performed a targeted lipidomic analysis of ex vivo-generated oxylipins from porcine aorta, coronary artery, pulmonary artery and perivascular adipose. Cyclooxygenase (COX)-derived prostanoids were the most abundant detectable oxylipin from all tissues. By contrast, the coronary artery produced significantly higher levels of oxylipins from CYP450 pathways than other tissues. The TLR4 ligand LPS induced prostanoid formation in all vascular tissue tested. The 11-HETE, 15-HETE, and 9-HODE were also induced by LPS from the aorta and pulmonary artery but not coronary artery. Epoxy fatty acid (EpFA) formation was largely unaffected by LPS. The pig CYP2J homologue CYP2J34 was expressed in porcine vascular tissue and primary coronary artery smooth muscle cells (pCASMCs) in culture. Treatment of pCASMCs with LPS induced a robust profile of pro-inflammatory target genes: *TNFα, ICAM-1, VCAM-1, MCP-1* and *CD40L*. The soluble epoxide hydrolase inhibitor TPPU, which prevents the breakdown of endogenous CYP-derived EpFAs, significantly suppressed LPS-induced inflammatory target genes. In conclusion, PPAR-activating oxylipins are produced and regulated in a vascular site-specific manner. The CYP450 pathway is highly active in the coronary artery and capable of providing anti-inflammatory oxylipins that prevent processes of inflammatory vascular disease progression.

## 1. Introduction

Peroxisome proliferator-activated receptors (PPARs) can be activated by a diverse group of endogenous fatty acid mediators including those produced from cyclooxygenase (COX), lipoxygenase and CYP450 enzymatic pathways [1]. These COX, lipoxygenase and CYP450 enzymes metabolize arachidonic acid and related polyunsaturated fatty acids, linoleic acid (LA), docosahexaenoic acid (DHA) and eicosapentaenoic acid (EPA) into series of biologically active oxylipin mediators [2,3,4]. Cyclooxygenases largely make prostanoids (and some hydroxyeicosatetraenoic acids (HETEs)) [5,6]. Lipoxygenases make hydroperoxy-eicostetraeoic acid HpETE, HETEs, hydroxyoctadecaenoic acids (HODEs), hydroxy-DHAs, and hydroxy-EPAs—some of which are the precursors for leukotrienes [2]. The PUFA-utilizing CYP450s metabolize fatty acids into series of oxylipin mediators through a combination of either epoxidation or lipoxygenase-like or ω- and ω-1-hydroxylation [2,3]. Using arachidonic acid as an example, CYP2J2 can produce both epoxyeicosatrienoic acids (EETs) and 19-HETE by its epoxygenase- and hydroxylase-like activities, respectively [3,7]. The PGD_2_ metabolite 15 deoxy-D^12,14^-PGJ_2_, PGI_2_, 8-, 12-, and 15-HETE, 9- and 13-HODE [1], and 8-, 9-, 11-, 12-, and 14–15-EET [8] have all been shown to activate PPARs. Soluble (sEH) and microsomal (mEH) epoxide hydrolases (EH; encoded by the gene ephx2 and ephx1 respectively) combine to metabolize nearly all EpFAs in vivo [9]. sEH inhibitors (sEH-I) inhibit the breakdown of EpFAs to their more soluble but less biologically active dihydroxy counterparts and potentiate EET signaling [10,11,12]. 

While several oxylipins can signal through known or yet-to-be-identified G-protein-coupled receptors, transient increases in oxylipin ligands can also induce PPAR activation toward a variety of downstream signals [1]. PPAR activation induces heterodimerization with other nuclear receptors such as the retinoid X receptor (RXR), which enhances binding to a consensus sequence (direct repeats of ‘AGGTCA’) referred to as PPAR response elements (PPREs). PPAR ligands have diverse roles in the cardiovascular system, from repression of genes encoding pro-inflammatory cytokines to induction (e.g., *TNFα*, IL1, IL6) of monocytes/macrophages toward foam cell morphology [13]. 

The roles of oxylipins are of long-standing interest in vascular biology [11,13,14,15,16,17,18,19,20]. COX products have both cardioprotective (prostacyclin; PGI_2_) and pro-thrombotic (e.g., thromboxane; TXA_2_) activity [20,21]. CYP450-derived EpFAs are anti-atherosclerotic, vasodilatory and anti-inflammatory [11,22,23,24,25,26,27,28,29,30], with the notable exception of LA-derived dihydroxyoctadecamonoenoic acids (DHOMEs), which regulate cardiac function [31], vascular development [32], and thermal hyperalgesia [33] at low levels, but are toxic at higher levels [34]. CYP450-derived EETs, in particular, were originally described in porcine coronary artery as an endothelium-derived hyperpolarizing factor produced in response to stimulation and stretch sEH-I-treated or sEH-knockout mice show protection to injury induced vascular neointima formation [25], atherosclerosis and aneurysm formation [26], hypertension [35,36], type 2 diabetes [37], and inflammatory cell recruitment [23,30]. Interestingly, in the pulmonary circulation, although sEH inhibitors have been shown to augment hypoxia-induced vasoconstriction, sEH inhibition or overexpression of EpFA-producing enzymes such as CYP2J2 is protective in various acute lung injury models [38,39,40]. 

We previously showed that PPARs can be activated by CYP2J2 and its products in vitro and in vivo [8]. A number of protective effects of CYP2J2 or EETs have now been shown to be mediated by PPARs, including the protective effects of laminar flow on endothelial cells [24], mediating coronary reactive hyperemia [41,42,43,44] and vascular response in soluble epoxide hydrolase-null mice [45], cytoprotection of cardiomyocytes [46], inhibition of angiotensin II cardiac remodeling [47] and abdominal aortic aneurysm formation [48], inhibition of renal interstitial fibrosis and inflammation [49], improved vascular function and decreased renal injury in hypertensive obese rats [50], and promoting angiogenesis and migration in human endothelial progenitor cells from acute myocardial infarction patients [51].

Pigs have a similar heart and cardiovascular system to humans and undergo spontaneous and diet-induced atherogenesis [52]. Here, we used a lipidomic approach to study endogenous oxylipin PPAR ligand production by the large vessels of the pig: the thoracic aorta compared to the coronary and pulmonary arteries. The vessel releasates were also compared to those of aortic perivascular adipose tissue ex vivo. Perivascular adipose was investigated as it has been shown to release various cytokines that act in an endocrine and paracrine manner to regulate vascular signaling and inflammation which have been implicated in the development of atherosclerosis, hypertension, neointimal formation, aneurysm, arterial formation and vasculitis [53,54]. 

Using a targeted lipidomic approach, we found coronary artery releases significantly more oxylipins of almost all classes than aorta and pulmonary artery. Perivascular adipose was a particularly rich source of COX-derived PGE_2_. Coronary artery was the highest source of CYP450-derived EpFAs PPAR ligands. The use of a sEH inhibitor TPPU on pig primary coronary artery vascular smooth muscle cells in culture showed strong anti-inflammatory activity consistent with PPAR activation. 

## 2. Materials and Methods

### 2.1. Materials

Authentic oxylipins (EETs, DHEQ, and HDPA) were from Cayman Chemical Company (Cambridge Bioscience, Cambridge, UK). SYBR green was from Takara. TPPU (*N*-[1-(1-oxopropyl)-4-piperidinyl]-N’-[4-(trifluoromethoxy)phenyl)-urea) was synthesized as previously described [55]. Unless stated, all other reagents were from Sigma-Aldrich (Poole, Dorset, UK).

### 2.2. Vessel Organ Culture

Abattoir pig vessels largely from white X female pigs aged 8–10 weeks old were obtained from the Royal Veterinary College. Fresh tissue was collected and used within 4 h. The 50–500 mg segments of vessel or perivascular aortic adventitia were cultured in serum-free DMEM supplemented with antibiotic/antimycotic mix (Sigma-Aldrich, St. Louis, MO., USA) at 37 °C, 5% CO2 and 95% air, as previously described for rat and human vessels [56,57]. Serum-free media was used, as most sera contain large amounts of oxylipins (unpublished observations). Organ culture was performed for just the first 24 h after explant in order to minimize cell differentiation. In some experiments, lipopolysaccharide (LPS; *E. coli*, 1 μg/mL) was given to induce an inflammatory response. 

### 2.3. Cell and Tissue Culture

Primary coronary artery smooth muscle cells (pCASMCs) were obtained by explant and grown as previously described for human vascular smooth muscle cells [58]. Briefly, extraneous tissue was removed, coronary arteries were opened along the midline, gently denuded, and chopped into small explants. SMCs were grown in DMEM supplemented with antibiotic/antimycotic mix, and 20% FBS, at 37 °C, 5% CO2 and 95% air. SMCs were identified by a characteristic morphological “hill-and-valley” growth pattern and by smooth muscle α-actin immunostaining. Since FBS interferes with lipid substrate composition and the release and detection of eicosanoids, all experiments were performed with DMEM supplemented with antibiotic/antimycotic mix and without FBS. 

### 2.4. Real-Time qRT-PCR

Pig CYP2J34, sEH, *TNFα*, *VCAM-1*, *ICAM-1*, *MCP-1* and *CD40* were measured using the SYBR Green ddCT method (see Appendix A for primer pairs). Targets were normalized to 18S expression. RNA was extracted using the ThermoScientific RNA extraction kit and 1 μg of total RNA was used to generate cDNA using Superscript II (Invitrogen) according to the manufacturer’s instructions. SYBR green qPCR was performed using Premix Ex Taq II mastermix (Takara) using a Chromo-4 machine and Opticon software. Genomic sequences were obtained from the UCSC Genome Browser website (http://genome.ucsc.edu/cgi-bin/hgGateway) and primers (see Appendix A) were designed from NCBI’s Primer Blast website (http://www.ncbi.nlm.nih.gov/tools/primer-blast/index.cgi?LINK_LOC=BlastHome).

### 2.5. Oxylipin Measurements

Explants were incubated in serum-free DMEM for 24 h, which allows for detection of both the highly abundant prostaglandins and HETEs and less-abundant CYP-derived oxylipins. LC–MS/MS analysis of oxylipin products in culture supernatants was as previously described [23,59]. LC–MS/MS analytes in samples were quantified against oxylipin standard curves (Cayman Chemical) using TraceFinder v4.1 (Thermo Scientific, Waltham, MA, USA) software.

### 2.6. Statistical Analyses

Graphical representations, heat maps and statistical analyses between groups (*t*-tests and paired *t*-tests) were performed using GraphPad Prism v8.1. When comparing multiple groups, ANOVA was followed by Holm–Sidak correction for multiple comparisons. All distributions appeared and were assumed to be normal. 

## 3. Results

### 3.1. Oxylipin Lipidomic Profiling of the Large Vessels of the Pig

Young female pigs were selected to be devoid of atherosclerosis and represent non-diseased vascular tissues. Fresh tissue explants were divided into various treatment groups and cultured in serum-free media for 24 h. Serum-free media was used, as most sera contain large amounts of oxylipins (unpublished observations). CYP-, LOX- and COX-derived oxylipins were detectable in organs culture for 24 h after explant. The most abundant oxylipin species represented in 24 h organ culture in all tissues were prostanoids derived from COX (Figure 1a). PGI_2_ was the major product from aorta (190 pg/mg) and pulmonary artery (640 pg/mg), whereas PGE_2_ was the major product from coronary artery (1135 pg/mg) and perivascular adipose (1390 pg/mg; Figure 1). The coronary artery generated by far the largest total amounts of measurable oxylipins followed by pulmonary artery, with the aorta producing approximately 1/8 of the prostanoids per unit weight as the coronary artery (Figure 1a). The coronary artery produced significantly more EpFA and hydroxy fatty acids than the aorta or perivascular adipose (Figure 1b; Appendix A), with the pulmonary artery production again intermediate between the aorta and coronary artery (Figure 1 and Figure 2). The perivascular adipose produced similar amounts of PGE_2_ as the coronary artery, with much lower relative levels of lipoxygenase or CYP450 products formed than any of the vessels (data not shown).

Interestingly, the aorta and coronary artery produced similar levels of COX products, with the notable exceptions of PGI_2_, which was significantly higher from aorta compared to coronary artery, and PGE_2_, which was higher in coronary artery compared to aorta (*p* < 0.05 unpaired *t*-test; Figure 1 and Figure 2). Lipoxygenase-derived HETEs and HODEs were also produced in significantly higher amounts by coronary artery than the aorta (Figure 3). In particular, LA-derived oxylipin epoxygenase and lipoxygenase products were produced at considerably higher levels (up to 90-fold) by coronary artery than aorta (Figure 1, Figure 2 and Figure 3). 

### 3.2. Regulation of Oxylipin Generation in the Large Vessels of the Pig by Inflammatory Stimuli: LPS/TLR4 Activation

Consistent with the well-established sensitivity of COX-2 induction, LPS elevated prostanoids in aorta, coronary artery, and pulmonary artery. Interestingly, LPS did not induce prostanoids in aortic perivascular adipose tissue (Figure 4a). In particular, the major vascular prostanoids PGI_2_ and PGE_2_ were significantly induced by LPS in vascular tissue (Figure 4). The 11-HETE, 15-HETE, 9-HODE and 13-HODE were significantly increased in the aorta and pulmonary artery, but not the coronary artery. With some exceptions, notably 19,20-EpDPE in pulmonary artery and 19-HETE in aorta (Figure 4a; Appendix A), LPS did not consistently alter lipoxygenase or CYP450 product levels (Figure 4a).

### 3.3. The sEH Inhibitor TPPU Reduces TLR-4 Induced Inflammation in pCASMCs

LPS did not induce the pig CYP2J homologue CYP2J34 in organ culture tissue (pulmonary artery and coronary artery) at 24 h or in primary pCASMCs (Figure 5a) at 4 h. By contrast, LPS strongly induced *TNFα* mRNA in both organ culture tissue and pCASMCs (Figure 5a). Although not induced by LPS, the endogenously produced EpFAs were anti-inflammatory in pCASMCs, as co-treatment of pCASMCs with the sEH-I TPPU (1 uM) significantly reduced LPS-induced *TNFα*, *ICAM-1*, *VCAM-1*, *MCP-1* (*CCL2*), and *CD40* mRNA (Figure 5b).

## 4. Discussion

We used a targeted lipidomic approach to identify the profile of oxylipins produced by pig coronary artery, aorta, pulmonary artery and aortic perivascular adipose tissue. In particular, the coronary artery was a major source of epoxygenase-derived oxylipins. Bovine and porcine coronary artery was one of the original sites for the discovery of vasoactive CYP450-derived EETs [60]. Here, we show that the coronary artery also produces CYP450 EpFAs from linoleic acid, EPA and DHA in significantly greater amounts than other large pig vessels. Outside of primates, the pig cardiovascular system is considered the most relevant to human biology. The pig heart is a similar size to human heart, and pigs can spontaneously undergo coronary artery disease [52]. The increased size of the pig compared to rodent models also means that it is also relatively easy to examine specific vascular responses in arteries such as the coronary artery, which would be very difficult in common rodent models.

Using fresh tissue in organ culture comes with certain caveats. Although directly comparative, these results are based upon 24 h accumulation of products. Our previous studies with organ culture indicate that fresh vessels are put under a mild inflammatory stress, which is associated with a low but significant level of COX-2 induction [56,57]. The results here are consistent with these previous studies [56,57]. Additionally, we know relatively little about the long-term stability of a number of oxylipins in media or biological fluids, but clearly there are differences. EETs for example are rapidly metabolized or reincorporated into membranes [3,61], so this 24 h accumulation analysis is likely to underestimate total EET production When we have examined acute oxylipin release (30 min) from rat aorta, prostaglandins and in particular PGI_2_ are still the most abundant species detected (DBB unpublished observation). Another caveat to this analysis is whether tissue-specific oxylipin metabolism is present. For example, coronary artery endothelial cells are known to metabolize EETs to chain-shortened epoxy-hexadecadienoic acids [62]; additionally, the presence of CYP4A3 may metabolize EETs into 20-OH derivative PPAR ligands [63], which were not included in our analysis.

LPS induces COX-derived prostanoids in rat and human vessels in organ culture in vitro [56,57]. All the pig vessels tested similarly produced prostanoids in response to LPS. The responsiveness of other oxylipin pathways to LPS is less well understood. Since activation of cPLA2 appears to be common to all three pathways, we hypothesized that lipoxygenase and CYP450 pathways would also be activated. Interestingly, the COX-derived eicosanoids were the only species commonly induced by LPS in all vessels. In aorta, but not coronary artery, 11-HETE and 15-HETE were similarly induced. The 11-HETE and 15-HETE are also potentially COX products [5], so it is intriguing why they are induced in the aorta and not coronary artery. Similar findings were previously found in human whole blood treated with LPS for 18 h [63]. Interestingly, HODEs were also induced by LPS in aorta, pulmonary artery and perivascular adipose tissue, which shows a selective induction of linoleic acid- and arachidonic acid-lipoxygenase pathways [64,65]. Unlike HETE induction, this HODE induction was not previously observed in human whole blood treated with LPS [66] but has been observed in the circulation of mice treated with LPS [67]. These lipidomic results clearly show a high compartmentalization between substrate generation and delivery to individual COX, lipoxygenease and CYP450 pathways. This data provides further impetus to look at the actions of these other oxylipin species. EPA and DHA are considered key components of the purported cardiovascular health benefits of oily fish. Supplementation of human or rodent diets with DHA and EPA increases DHA and EPA EpFAs [68,69]. Coronary artery metabolism of EPA and DHA into EpFAs could therefore contribute to these dietary lifestyle modifications in cardiovascular disease. Additionally, further investigations are required to understand the role of PPAR signaling in any effects.

The coronary artery is of particular interest for vascular research, since coronary artery disease and occlusion is the major cause of heart attacks in humans. There has been considerable interest in both the potential cardioprotective effects of PPAR ligands and testing sEH inhibitors in cardiovascular disease [25,26]. As recently reviewed PPAR ligands in experimental animal models have all shown to reduce aortic atherosclerosis [70,71]. There has been considerable interest in whether these findings translate into humans [70,71]. Both PPARα and PPARγ agonists have shown some mild clinical efficacy in reducing cardiovascular event [71]. However, the clinical efficacy of the PPARγ ligand rosiglitazone has been questioned as it appeared to increase cardiovascular events in an early trial [71]. Nonetheless, there has been considerably recent interest in developing selective modulators, and dual- and pan-PPAR agonists, that have increased efficacy and reduced side effects [70,71]. We hypothesize that the potential endogenous oxylipin PPAR ligands are more likely to act as pan/dual or selective modulator-type agonists. sEH inhibitors are anti-hypertensive, anti-diabetic, anti-obesity and reduce the development of aortic atherosclerosis in mouse models [25,26,59,72,73]. Importantly, atherosclerosis is rarely investigated in the coronary circulation of mice. More often, aortic atherosclerotic plaque formation is used as a surrogate for coronary artery disease. Our oxylipin profiling suggests that aorta CYP activity underestimates that found in coronary arteries and likely underpredicts the role of CYP-derived oxylipins in coronary artery atherosclerosis. Preservation of the coronary circulation therefore may be so critical that it has evolved this higher EpFA system to maintain flow and limit inflammation. Originally, these positive benefits of sEH inhibitors were attributed to lipid-lowering actions [26]. Human coronary artery disease, in particular obstructive coronary artery disease, is associated with decreased circulating EETs [74,75]. In the heart and coronary circulation, CYP2J2 or EETs mediate coronary reactive hyperemia [41,42,43,44], cytoprotection of cardiomyocytes [46], and inhibition of angiotensin II cardiac remodeling [47] in part by the activation of PPARs. sEH inhibition acts to maintain higher levels of EETs/EpFAs or may shunt to alternative PPAR ligands such as the 20-OH CYP4A derivatives. Although, DHETs are also PPAR activators [76], the concentrations required are 10–100-fold higher than published for EETs [8,64]; thus, sEH inhibition will act to promote the PPAR agonist activity of CYP-derived epoxides. 

The coronary artery was the largest source of epoxygenase products of all the major vessels we tested, suggesting that multiple oxylipin species may have particularly important roles at this site. The pig homologue of human CYP2J2 is CYP2J34 [77]. In human monocytes and endothelial cells, we found that LPS induced CYP2J2 [28,29,78]. By contrast, LPS did not induce CYP2J34 in pig vessels or monocytes (Figure 5a, DBB unpublished observations), indicating at least one difference between human and pig CYP2J enzymes. We previously reported differences in intimal and medial SMC phenotypes isolated from the rat [79,80]. Medial SMCs but not intimal SMCs were sensitive to the anti-inflammatory actions of sEH inhibitors [81]. The sEH inhibitor TPPU inhibited inflammatory mediators induced by TLR-4 activation in primary pCASMCs. This is the first time anti-inflammatory actions have been described in coronary tissue for sEH inhibitors, and this further supports an anti-inflammatory/pro-resolution role for the sEH pathway in mediating cardioprotective actions. The pCASMCs we cultured represent a classical medial SMC phenotype, with a classical spindle shape and hill-and-valley morphology. No epithelial cell types were observed in these primary cultures. We have yet to determine whether distinct porcine ‘intimal’ SMC phenotypes can be identified that share these different properties. 

The aorta produced the lowest levels of oxylipins, with the pulmonary artery in between the aorta and coronary artery. The lack of activity in the aorta may just reflect the aorta’s main role as a conduit vessel and one not particularly responsive to vasoactive mediators or a major site for human vascular disease initiation. The pulmonary artery produced the highest levels of basal and LPS-inducible PGI_2_, consistent with the importance of this eicosanoid in maintaining pulmonary health [79], which may be in part mediated by activation of PPARβ/δ (or PPARα) [1,82]. CYP450-derived eicosanoids contribute to hypoxia-induced pulmonary hypertension [83] and are protective in models of inflammation [38,39,40,84]. The production of these oxylipin mediators from the pulmonary artery suggest that the pig may also be a useful translational model to study oxylipins and PPARs on pulmonary health and disease. Here, we show perivascular adipose is also a large potential source of oxylipins, in particular PGE_2_, and further suggest a role for oxylipins from alternative cellular sources as potential mediators of vascular health and disease. Similarly, the relative contribution of vascular cell types—endothelial, smooth muscle phenotypes, adventitial fibroblasts and adipose—require further investigation.

## 5. Conclusions

We have performed a lipidomic analysis on large vessels and perivascular adipose from the pig. Although prostanoids were the dominant detectable species from all tissue, the coronary artery produced considerably more oxylipins in terms of species and amounts when compared to the aorta and pulmonary artery, in particular those from CYP450 pathways. Using porcine pCASMCs, we showed using the sEH inhibitor TPPU that endogenous CYP-derived epoxy-oxylipin PPAR ligands were strongly anti-inflammatory. The CYP450 pathway in the coronary artery not only provides vasodilator tone, but here we propose an anti-inflammatory tone that helps to prevent processes of vascular disease progression. These results also further highlight the potential for sEH inhibitors as therapies for cardiovascular and inflammatory diseases.

## Figures and Tables

**Figure 1 cells-09-01096-f001:**
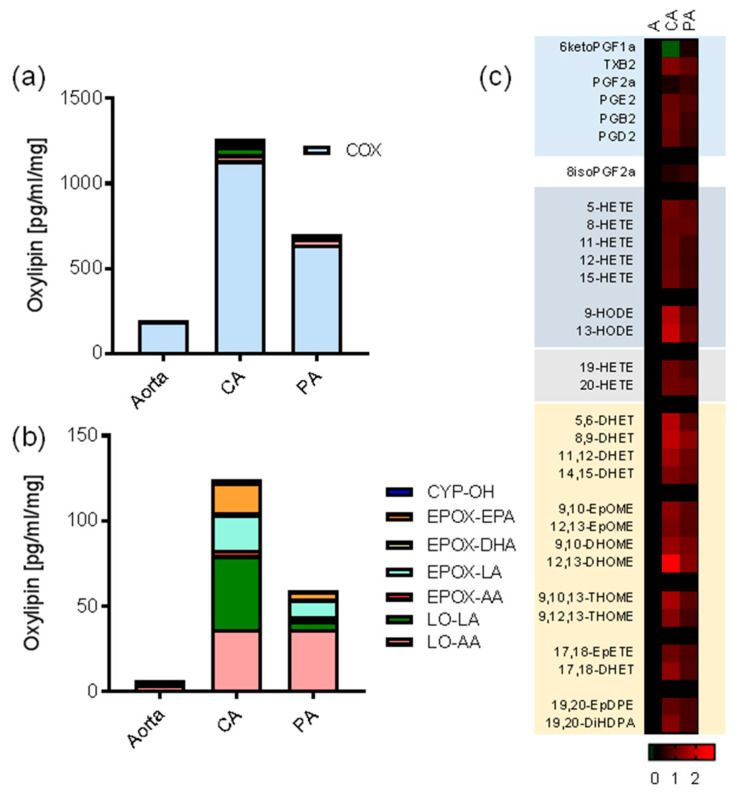
Characterization of oxylipin production from aorta, coronary artery, and pulmonary artery. (**a**,**b**) Comparative and relative contribution of cyclooxygenase (COX), lipoxygenase-arachidonic acid (LO-AA), lipoxygenase-linoleic acid (LO-LA), CYP-epoxygenase-arachidonic acid (EPOX-AA), CYP-epoxygenase-linoleic acid (EPOX-LA), CYP-epoxygenase-DHA (EPOX-DHA), CYP-epoxygenase-EPA (EPOX-EPA), and CYP-ω-hydroxylase (CYP-OH) products to the oxylipin releasate of aorta, coronary artery (CA), and pulmonary artery (PA) in 24 h organ culture. (**a**) shows all pathways, whereas (**b**) shows all pathways minus COX. Bars are based upon the single most oxylipin abundant oxylipin product detected in each pathway which is used as a representative index of oxylipin class. (**c**) Heatmap showing Log10 fold differences in the mean amount of each oxylipin detected from coronary artery (CA) and pulmonary artery (PA) compared to aorta. The actual fold range in the coronary artery was 0.5-fold for 6-keto PGF_1α_ to 823-fold for 12,13-DHOME. Data represents organ culture from *n* = 3–4 separate animals.

**Figure 2 cells-09-01096-f002:**
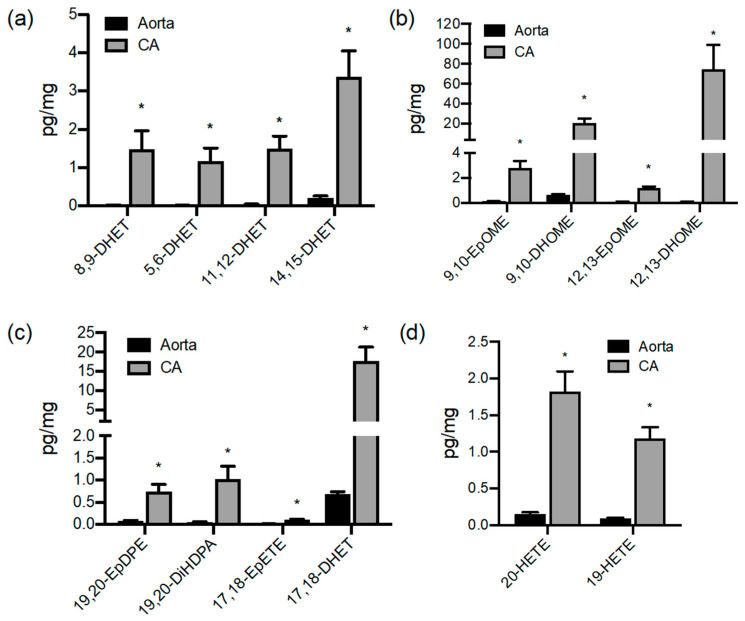
Coronary arteries produce high levels of CYP-derived oxylipins. Figures show detectable CYP epoxygenase (**a**) EPOX-AA, (**b**) EPOX-LA, (**c**) EPOX-DHA/EPA and (**d**) CYP-OH products released by pig aorta (black bars) and coronary artery (grey bars). Oxylipins accumulated in 24 h serum-free organ culture were measured by LC–MS/MS and expressed as pg/mg of wet tissue weight. Data represents organ culture from *n* = 3–4 separate animals. Data represents organ culture from *n* = 3–4 separate animals. * indicates *p* < 0.05 between Aorta and CA.

**Figure 3 cells-09-01096-f003:**
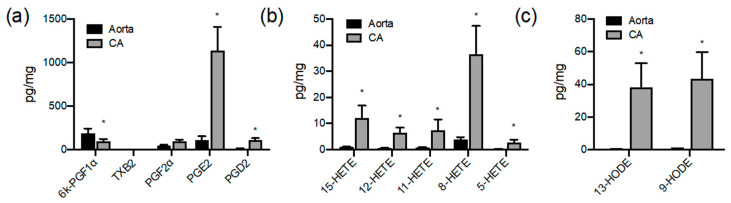
Comparison of aortic and coronary artery production of cyclooxygenase and ‘lipoxygenase’ oxylipin products. Figures show (**a**) cyclooxygenase, (**b**) ‘lipoxygenase’ products of arachidonic acid (HETEs) and (**c**) linoleic acid (HODEs) products released by pig aorta (black bars) and coronary artery (grey bars). Oxylipins accumulated in 24 h serum-free organ culture were measured by LC–MS/MS and expressed as pg/mg of wet tissue weight. Data represents organ culture from *n* = 3–4 separate animals. * indicates *p* < 0.05 between Aorta and CA.

**Figure 4 cells-09-01096-f004:**
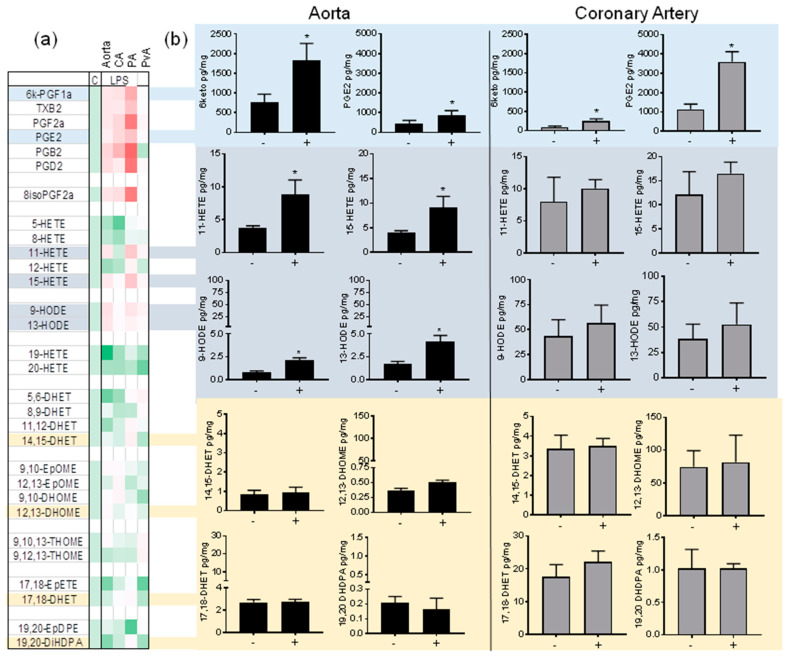
Regulation of oxylipin production in large vessels by LPS/TLR4 activation. (**a**) Heatmap showing summary of fold differences in the mean oxylipin generation in aorta, coronary artery (CA), pulmonary artery (PA) and perivascular adipose (PvA) untreated tissue (C) compared to tissue treated with LPS (1 μg/mL) ex vivo. The range of fold differences was from 0.5- (19-HETE; Aorta) to 9-fold (PGB_2_; PVA). (**b**) Comparison of major oxylipin production: 6-ketoPGF_1α_, PGE_2_, 11-HETE, 15-HETE, 9-HODE, 13-HODE, 14,15-DHET, 12,13-DHOME, 17,18-DHET and 19,20-DHDPA in aorta and coronary artery treated in the absence (-) or presence regulation by LPS (1 μg/mL; +). * indicates *p* < 0.05 by unpaired *t*-test between tissue treated in the presence of absence of LPS. Data represents organ culture from *n* = 3–4 separate animals.

**Figure 5 cells-09-01096-f005:**
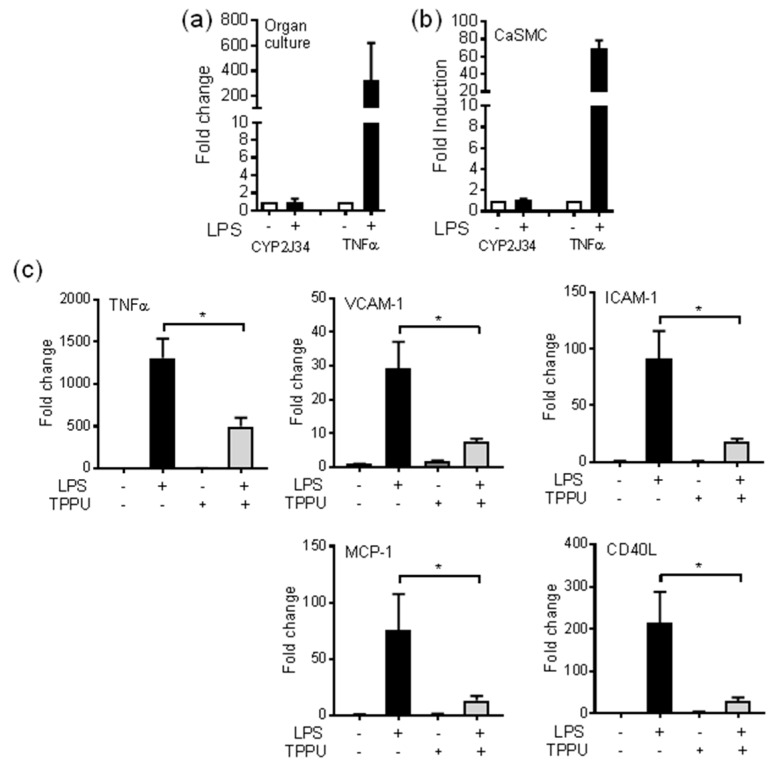
The sEH inhibitor TPPU is anti-inflammatory in coronary artery vascular smooth muscle cells. Expression of TNFα and CYP2J34 mRNA in (**a**) combined pig coronary and pulmonary artery vessels in organ culture at 24 h (*n* = 4), and (**b**) pig primary coronary artery cells at 4 h (CaSMCs) in the presence or absence of LPS (1 μg/mL). mRNA was measured by qRT-PCR and fold levels normalized to 18S. (**c**) Inflammatory target gene expression of TNFα, VCAM-1, ICAM-1, MCP-1 (*CCL2*) and *CD40* in cultures of pCASMCs in the presence or absence LPS (1 μg/mL; 4 h), and/or sEH inhibitor TPPU (1 μM; given as a 1 h pretreatment before addition of LPS). * indicates *p* < 0.05 by paired *t*-test between cells treated with TPPU in the presence of absence of LPS. Data represents mean ± SE from *n* = 4 cultures from two separate animals.

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
