# Peer review of "Vascular Lipidomic Profiling of Potential Endogenous Fatty Acid PPAR Ligands Reveals the Coronary Artery as Major Producer of CYP450-Derived Epoxy Fatty Acids"

_cells, 2020, doi:10.3390/cells9051096_

Round 1

Reviewer 1 Report

Overall, this is a nicely written article. However, I do have some concerns which need to be addressed

ABSTRACT:

The abstract is quite unstructured and should follow the pattern introduction>hypothesis>methods>results>conclusion

INTRODUCTION:

The introduction is nicely written while somewhat fragmented. I suggest the authors add some additional background to provide a better foundation for the described signaling pathways. Do not ever assume that all your readers have your level of knowledge. 

MATERIALS & METHODS

Why where female pigs used? Younger human females are normally protected from atherogenesis by estrogen. If this is not the case in pigs that should be stated. If pigs do have the same phenomenon a better justification needs to be give as to why this experimental model was selected. 

Why where the animals so young? Vessels from 8 week pigs will still be in their developmental phase which means that the smooth muscle cells have not reached their full differentiation. This is a huge issue as the authors claim to study the effects of atherogenesis while the model used will not allow for full phenotype shift. 

How was the smooth muscle cells isolated for these experiments? Where the vessels denuded? What was done to remove the pericytes and other adventitial cells? 

Were the cell cultures done in serum free media? Why/why not? Please justify the experimental design

It should be stated in this section which vessels were used and why. Atherosclerosis does not develop in all vessels (for example not in pulmonary vessels) so the selection needs to be justified. 

There needs to be a section for 'statistics' in the 'Materials & methods' that describes the tests that were used as well as how the authors tested for normal distribution. 

RESULTS

What was the purpose of organ culture for 24h? Justify!

Was only one experiment performed for the data described in Figure 1 or did the authors forget to include error bars? 

Figure 2: This is a terrible figure as the values for the aorta are impossible to interpret. Redo and make it right!

Same comment for Figure 3

What were the control values (before culture)? How can the readers know that these values were not present before the culture. Especially considering the poor choice of experimental model (see above). 

The statistical test (t-test) for figure 5 is incorrect as there are more than one variable. This comparison should have been done by an ANOVA after control of normal distribution. 

Data displayed is not mean±sem but rather mean±se

DISCUSSION

How can the authors draw these conclusions when they have no control tissue for relative comparison as well as a very, very low n-value? 

What is the basis for the authors claim that there is a potential for sEH inhibitors for therapy of CVDs when they did not perform any tests for CVD or smooth muscle dedifferentiation? 

Overall, the authors need to better justify the reasons why this study was performed. 

Author Response

Overall, this is a nicely written article. However, I do have some concerns which need to be addressed

We thank the reviewer for their careful reading of the manuscript and positive feedback.

ABSTRACT:

The abstract is quite unstructured and should follow the pattern introduction>hypothesis>methods>results>conclusion

Thank you for the suggestion. The abstract was in most need of several more sentences of introduction and hypothesis to set up the rationale for the study. “The very short biological half-lives of oxylipins suggest roles as autocrine or paracrine signaling molecules. Improving our understanding of the compartmentalized origins of oxylipin PPAR ligands will increase our knowledge of PPAR signalling in diverse vascular tissues.” (Lines 16-21)

INTRODUCTION:

The introduction is nicely written while somewhat fragmented. I suggest the authors add some additional background to provide a better foundation for the described signaling pathways. Do not ever assume that all your readers have your level of knowledge. 

Additional background has been added to the introduction: “While several oxylipins can signal through known or yet-to-be-identified G-protein-coupled receptors, transient increases oxylipin ligands can induce PPAR activation toward a variety of downstream signals [1]. PPAR activation induces heterodimerization with other nuclear receptors such as the retinoid X receptor (RXR) which enhance binding to a consensus sequence (direct repeats of ‘AGGTCA’) referred to as PPAR response elements (PPREs). PPAR ligands have diverse roles in the cardiovascular system, from repression of genes encoding pro-inflammatory cytokines to induction of monocytes/macrophages toward foam cell morphology [13].” (Lines 57-63)  

MATERIALS & METHODS

Why where female pigs used? Younger human females are normally protected from atherogenesis by estrogen. If this is not the case in pigs that should be stated. If pigs do have the same phenomenon a better justification needs to be give as to why this experimental model was selected. 

We added this statement to the beginning of the results (Lines 146-7): “Young female pigs were selected to be devoid of atherosclerosis and represent non-diseased tissues.” As this is our first study on large pig vessels we chose fresh abattoir tissue as the most ethical source of tissue. Though we absolutely agree this would be an interesting and important comparison, in the commercial setting male pigs are not commercially bred for food, only very small numbers are kept and used solely for breeding.

Why where the animals so young? Vessels from 8 week pigs will still be in their developmental phase which means that the smooth muscle cells have not reached their full differentiation. This is a huge issue as the authors claim to study the effects of atherogenesis while the model used will not allow for full phenotype shift. 

This manuscript describes a comprehensive analysis which shows substantial difference in oxylipins production between vascular beds and adipose. A comparison of normal vs atherogenic tissues would be a separate study investigated with different endpoints. These tissues were from the abattoir and the animals represent the usual age for the commercial use of pigs in meat production. Interestingly, we have identified early athero-lesions in aorta’s of female pigs of this age from rare breed pigs (but not the ‘commercial’ breed used in this study). Ideally, we plan future comparisons of these animals and tissue.

How was the smooth muscle cells isolated for these experiments? Where the vessels denuded? What was done to remove the pericytes and other adventitial cells? 

Essentially yes, the vessels are large enough to obtain medial tissue, and were cleaned of adventita and denuded. We have expanded the Method’s section on SMC isolation to read: “Briefly, extraneous tissue was removed, coronary arteries were opened along the midline, gently denuded, and chopped into small explants. SMC were grown in DMEM supplemented with antibiotic/antimycotic mix, and 20% FBS; 37°C; 5% CO2; 95% air. SMCs were identified by characteristic morphological “hill-and-valley” growth pattern and by smooth muscle α-actin immunostaining.” (Lines 117-21)

Were the cell cultures done in serum free media? Why/why not? Please justify the experimental design.

This sentence was added to the Methods (line x) “Serum-free media was used as most serum contains large amounts of oxylipins (unpublished observations).” In addition, most commercial serum lots also contain traces of the enzymes that degrade both prostaglandins and epoxy-fatty acids.

It should be stated in this section which vessels were used and why. Atherosclerosis does not develop in all vessels (for example not in pulmonary vessels) so the selection needs to be justified. 

There needs to be a section for 'statistics' in the 'Materials & methods' that describes the tests that were used as well as how the authors tested for normal distribution. 

This is a good point and one of the reasons we wanted to make these initial comparisons added a statistical section to the Methods: “Graphical representations, heat maps and statistical analyses between groups (ttests and paired ttests) were performed using GraphPad Prism v 8.1. When comparing multiple groups, ANOVA was followed by Holm-Sidak correction for multiple comparisons. All distributions appeared were assumed to be normal.” Normal distributions. Statistical analyses were not improved by either log- nor square root-transformation of the LC/MS/MS data.

RESULTS

What was the purpose of organ culture for 24h? Justify!

The 24-hour timepoint was chosen as a compromise. It allowed for detection of prostaglandins and HETEs (these would have been detectable at earlier timepoints), and lesser abundant CYP-derived oxylipins, which in our experience are poorly detectable in unstimulated cells at early timepoints. This has been explained in the Methods section “Explants were incubated in serum free DMEM for 24 hours, which allows for detection of both the highly abundant prostaglandins and HETEs and less abundant CYP-derived oxylipins.”

Was only one experiment performed for the data described in Figure 1 or did the authors forget to include error bars? 

This figure displays several significant points: First, by weight, coronary and pulmonary artery produce significantly more total oxylipins than aorta. Second, most oxylipins produced are prostaglandins. Third, there are striking differences in CYP- and LOX-derived oxylipins in aorta vs CA and adipose. This was not an N=1 experiment. Error bars could be added for each color/analyte group on the figure; however, we believe that projection muddle the plot substantially. It is our attempt to graphically represent the data in a simple summary form.

Figure 2: This is a terrible figure as the values for the aorta are impossible to interpret. Redo and make it right! Same comment for Figure 3

The figure was presented as it was (note equal y-axis scaling for A and B) to highlight the differences between vascular beds. In response to the reviewer’s comment, we have plotted the data side by side in the same graph. See the new Figure 2 and Figure 3.

What were the control values (before culture)? How can the readers know that these values were not present before the culture. Especially considering the poor choice of experimental model (see above). 

It is unclear to which experiment this comment refers. In the case of Figures 2 and 3, the explants were washed and placed into serum free media which contains no oxylipins. The experiment was to determine the amount of oxylipins produced by basal and stimulated tissues.

The statistical test (t-test) for figure 5 is incorrect as there are more than one variable. This comparison should have been done by an ANOVA after control of normal distribution. 

Data displayed is not mean±sem but rather mean±se.

As suggested, a new paragraph was added to the Methods: “Graphical representations, heat maps and statistical analyses between groups (ttests and paired ttests) were performed using GraphPad Prism v8.1. When comparing multiple groups, ANOVA was followed by Holm-Sidak correction for multiple comparisons. All distributions appeared and were assumed to be normal.” (Line 141-4)

DISCUSSION

How can the authors draw these conclusions when they have no control tissue for relative comparison as well as a very, very low n-value? 

What is the basis for the authors claim that there is a potential for sEH inhibitors for therapy of CVDs when they did not perform any tests for CVD or smooth muscle dedifferentiation? 

Overall, the authors need to better justify the reasons why this study was performed. 

The reviewer raises several disparate points. Additional statements were added to the manuscript to justify the rationale for this investigation (differential PPAR ligand production in various vascular beds). While the reviewer suggests that there was no control tissue, the point was to show the varied oxylipin profiles in aorta, coronary artery, pulmonary artery and perivascular adipose. The anti-inflammatory actions of the sEH inhibitor support a immunomodulatory role for these PPAR ligands in atherogenesis. As the effects of PPAR ligands can be have both pro and anti-atherogenic effects, it remains to be seen whether these sEH inhibitors will have a net benefit in atherosclerosis. Our data suggests that CYP-derived oxylipins are poorly represented in aorta but more abundant in coronary arteries. While coronary arterial lesions lead to most myocardial infarctions, aortic lesions are used as a surrogate in animal models. Our data suggests that aorta is a poor surrogate for understanding the role of CYP-derived oxylipins in coronary arterial atherosclerosis. Additional sentences in the abstract and discussion have been added to clarify the hypothesis and conclusions of this study. (Lines 16-21, 282-284)

Reviewer 2 Report

The authors employed targeted lipidomic analysis to profile oxylipins produced by pigs’ large vessels. The authors found that primary coronary artery cells produced higher levels of oxylipins from CYP450 pathways than other tissues. Endogenously produced epoxy-fatty acids are ligands of PPARs and were anti-inflammatory in primary coronary artery cells. While experiments are well designed and correctly performed, manuscript is not easy to read.

Here are several shortfalls:

  • The manuscript contains a large number of abbreviations; however, it does not have a list of used acronyms. Lack of the list of abbreviations makes it difficult to read or review the manuscript.
  • Statement(s) and references highlighting the role of PPARs in anti-inflammatory response should be added.
  • Information in Methods is very sparse. More details are needed. Were bioinformatic tools used? How heatmaps were generated?
  • Figure 2. Second and third graphs in the Aorta panel do not contain any visible data. It would be more informative if values of each datapoint are indicated at the top of each graph. Broken axis graphs also can be used. The same applies to Figure 3.
  • Are there any other possible functions of oxylipins produced by large vessels, besides providing an anti-inflammatory tone? These possible functions of PPAR pathway should be discussed in more detail.
  • Mass-spectra of key LC/MS experiments should be provided in the supplemental file.

Author Response

The authors employed targeted lipidomic analysis to profile oxylipins produced by pigs’ large vessels. The authors found that primary coronary artery cells produced higher levels of oxylipins from CYP450 pathways than other tissues. Endogenously produced epoxy-fatty acids are ligands of PPARs and were anti-inflammatory in primary coronary artery cells. While experiments are well designed and correctly performed, manuscript is not easy to read.

Here are several shortfalls:

The manuscript contains a large number of abbreviations; however, it does not have a list of used acronyms. Lack of the list of abbreviations makes it difficult to read or review the manuscript.

We prepared a list of uncommon abbreviations used. (Lines 333-41)

Statement(s) and references highlighting the role of PPARs in anti-inflammatory response should be added.

Both reviewers 1 and 3 had similar requests. Additional statements on the role of PPARs in anti-inflammation (Lines 57-63) 

Information in Methods is very sparse. More details are needed. Were bioinformatic tools used? How heatmaps were generated?

Many additional clarifications were added to the methods (Lines 112-13, 117-21, 135-39, 141-44) Briefly, heatmaps were generated using GraphPad Prism. We generated and use an extensive automated spreadsheet to calculate analyte concentrations but no extensive ‘bioinformatical’ approaches were used in this manuscript.

Figure 2. Second and third graphs in the Aorta panel do not contain any visible data. It would be more informative if values of each datapoint are indicated at the top of each graph. Broken axis graphs also can be used. The same applies to Figure 3.

This was also noted by reviewer 2. These graphs were merged to highlight the differences between aorta and coronary artery.

Are there any other possible functions of oxylipins produced by large vessels, besides providing an anti-inflammatory tone? These possible functions of PPAR pathway should be discussed in more detail.

A number of these points are discussed in the introduction including vascular tone, angiogenesis, cellular migration, cytoprotection and fibrosis (lines 77-85). We are happy to be guided by the reviewer to other points that might be discussed.

Mass-spectra of key LC/MS experiments should be provided in the supplemental file.

We do have these but the data is on computers that are unexpectedly inaccessible due to the coronavirus lockdown. We have published extensively with this method and are happy to add at these in as a supplement at future date when they become available.

Reviewer 3 Report

This manuscript studied lipidomic profiles of ex vivo incubated pig aorta, coronary artery, pulmonary artery and perivascular adipose.  The authors have shown that coronary artery is a major producer of CYP450-derived oxylipins compared with other tissues examined.  They have also reported LPS-induced upregualtion of pro-inflammatory genes in ex vivo cultured tissues (coronary artery and pulmonary artery) and cells (pCASMC), which can be blocked by treatment of TPPU. 

Major: 

  1. What's the mechanism of EpFAs-dependent LPS-induced upreglation of pro-inflammatory genes, while EpFAs were not induced by LPS? Since LPS did not alter CYP2J34 in organ culture, SMC or monocytes, how about pig ECs? The same group has shown that LPS increased CYP2J2 in human ECs. Is EC-SMC crosstalk involved in the process of LPS-induced upreglation of pro-inflammatory genes? 

Minor:

  1. Fig1b, the legend is too small to tell the colors. 
  2. Line 124, Fig1 didn't show data from perivascular adipose. 
  3. Line 264, reference 78, wrong reference.

Author Response

This manuscript studied lipidomic profiles of ex vivo incubated pig aorta, coronary artery, pulmonary artery and perivascular adipose.  The authors have shown that coronary artery is a major producer of CYP450-derived oxylipins compared with other tissues examined.  They have also reported LPS-induced upregualtion of pro-inflammatory genes in ex vivo cultured tissues (coronary artery and pulmonary artery) and cells (pCASMC), which can be blocked by treatment of TPPU.

We thank the reviewer for their time and constructive feedback

Major:

What's the mechanism of EpFAs-dependent LPS-induced upreglation of pro-inflammatory genes, while EpFAs were not induced by LPS?

LPS is well-known to induce cytokines and COX-2 induction through TLR4 and NF-kb-mediated signaling. Induction of prostaglandins is likely due to upregulation of COX-2.  This has been extremely well investigated by many others. EpFAs are not induced in our system. Unlike previous studies in monocytes, CYP2J is not induced in isolated organ cultures or SMCs (Figure 5). 

Since LPS did not alter CYP2J34 in organ culture, SMC or monocytes, how about pig ECs? The same group has shown that LPS increased CYP2J2 in human ECs. Is EC-SMC crosstalk involved in the process of LPS-induced upreglation of pro-inflammatory genes?

The reviewer raises several interesting questions. We have not yet looked at the ECs, but suspect CYP2J34 is just not inducible by LPS as we see with human CYP2J2. We suspect, but have no data to confirm, that the cytokines are produced by both endothelial cells and SMC in response to LPS; we have seen this in human denuded vessels in organ culture previously. Among the blood vessels cell types, ECs are believed to be the major EpFA producers, which have paracrine activities on SMCs. The effect of EpFAs on EC activation has been extensively studied elsewhere (eg Deng, FASEB 2011). The point of these studies was to examine differences between vascular beds in and ex vivo setting that retained relevant tissue architecture and cell/cell interactions.

Minor:

Fig1b, the legend is too small to tell the colors. The legend has been expanded to show colors.

Line 124, Fig1 didn't show data from perivascular adipose. PvA has been removed from this legend

Line 264, reference 78, wrong reference. The discord in references above # X has been corrected

Round 2

Reviewer 1 Report

The authors have responded to my previous concerns in an acceptable manner. 

Reviewer 2 Report

Authors addressed almost all my concerns. It can be published in the current form.

Reviewer 3 Report

The authors have addressed my previous concerns. I don't have further questions.